| 1  | On the Nationwide Variability of Low-Level Jets Prior                                                                                            |  |  |  |  |  |
|----|--------------------------------------------------------------------------------------------------------------------------------------------------|--|--|--|--|--|
| 2  | to Warm-season Nocturnal Rainfall in China Revealed                                                                                              |  |  |  |  |  |
| 3  | by Radar Wind Profilers                                                                                                                          |  |  |  |  |  |
| 4  | Ning Li <sup>a,b</sup> , Jianping Guo <sup>a,d*</sup> , Xiaoran Guo <sup>a,d</sup> , Tianmeng Chen <sup>a,d</sup> , Zhen Zhang <sup>a</sup> , Na |  |  |  |  |  |
| 5  | Tang <sup>a</sup> , Yifei Wang <sup>a</sup> , Honglong Yang <sup>d</sup> , Yongguang Zheng <sup>c</sup>                                          |  |  |  |  |  |
| 6  |                                                                                                                                                  |  |  |  |  |  |
| 7  | <sup>a</sup> State Key Laboratory of Severe Weather Meteorological Science and Technology &                                                      |  |  |  |  |  |
| 8  | Specialized Meteorological Support Technology Research Center, Chinese Academy                                                                   |  |  |  |  |  |
| 9  | 9 of Meteorological Sciences, Beijing 100081, China                                                                                              |  |  |  |  |  |
| 10 | <sup>b</sup> College of Earth and Planetary Sciences, University of Chinese Academy of                                                           |  |  |  |  |  |
| 11 | Sciences, Beijing 100049, China                                                                                                                  |  |  |  |  |  |
| 12 | <sup>c</sup> National Meteorological Centre, Beijing 100081, China                                                                               |  |  |  |  |  |
| 13 | <sup>d</sup> CMA Field Scientific Experiment Base for Low-Altitude Economy Meteorological                                                        |  |  |  |  |  |
| 14 | Support of Unmanned Aviation in Guangdong-Hong Kong-Macao Greater Bay Area,                                                                      |  |  |  |  |  |
| 15 | Shenzhen 518108, China                                                                                                                           |  |  |  |  |  |
| 16 |                                                                                                                                                  |  |  |  |  |  |
| 17 |                                                                                                                                                  |  |  |  |  |  |
| 18 |                                                                                                                                                  |  |  |  |  |  |
| 19 | Correspondence to:                                                                                                                               |  |  |  |  |  |
| 20 | Dr./Prof. Jianping Guo (Email: jpguocams@gmail.com)                                                                                              |  |  |  |  |  |
| 21 |                                                                                                                                                  |  |  |  |  |  |

https://doi.org/10.5194/egusphere-2025-4939 Preprint. Discussion started: 16 October 2025 © Author(s) 2025. CC BY 4.0 License.

22 **Abstract** 23 Nocturnal rainfall initiation is closely linked to low-level jets (LLJs), but national-scale 24 LLJ features over China—especially their evolution preceding warm-seasonal 25 nocturnal rainfall-remain unknown due to scarce high-resolution vertical wind 26 observations. Here, we reveal the multiscale responses of LLJs within 2 hours 27 preceding the onset of nocturnal heavy rain (HR) and non-HR across four phases of 28 rainy seasons in China during the warm season (April-October) of 2023–2024, utilizing 29 data from a nationwide network of radar wind profilers (RWPs) in combination with 30 surface observations and reanalysis data. Results show that nocturnal rainfall accounted 31 for over 50% of warm-season rainfall, with 56% preceded by LLJs within 2 hours 32 leading up to their onset. In monsoon regions, ~40% of nocturnal HR (LLJ HR) were 33 LLJ-associated and had higher heavy rainfall probability than non-LLJ HR. Critically, 34 these LLJ HR events depended on rapid, coupled minute-scale dynamical 35 intensification, typically occurring 30 to 120 minutes before rainfall initiation. 36 Specifically, coherent changes in LLJ wind profiles, frequency, vertical wind shear, and 37 pronounced evolution in jet height were observed, operating in synergy with substantial 38 thermodynamic instability. This behavior stood in sharp contrast to LLJ non-HR 39 events, which were characterized by relatively stable, weakening, or declining LLJ 40 evolution and an inadequate thermodynamic response. These findings demonstrate that 41 a minute-scale 'rapid reorganization' of dynamic and thermodynamic conditions driven 42 by swift evolution of the LLJ is essential for nocturnal HR, providing observational 43 constraints for regional model parameterizations and nowcasting. 44

https://doi.org/10.5194/egusphere-2025-4939 Preprint. Discussion started: 16 October 2025 © Author(s) 2025. CC BY 4.0 License.

| 45 |                                                                                            |
|----|--------------------------------------------------------------------------------------------|
| 46 | Short Summary                                                                              |
| 47 | Nighttime rainfall often links to low-level jets (LLJs), but we lack clarity on nationwide |
| 48 | LLJ features due to limited wind data. To address this, we used a nationwide radar wind    |
| 49 | profiler network to study LLJ changes 2 hours before rainfall, covering China's 2023-      |
| 50 | 2024 rainy seasons. 56% nighttime rainfall had LLJs. The heavy rains linked to LLJs        |
| 51 | needed LLJs to strengthen quickly 30-120 minutes preceding rain. These findings show       |
| 52 | the importance of LLJ in nowcasting nighttime rainfall.                                    |
| 53 |                                                                                            |
| 54 |                                                                                            |
| 55 |                                                                                            |
| 56 |                                                                                            |
| 57 |                                                                                            |
| 58 |                                                                                            |
| 59 |                                                                                            |
| 60 |                                                                                            |

### 1. Introduction

62 Forecasting nocturnal heavy rainfall (HR) and associated severe convective 63 weather remains a major challenge in hazardous weather prediction (Davis et al., 2003; 64 Trier et al., 2006), owing to the complexity of triggering mechanisms, the scarcity of 65 continuous high-resolution observations, and inaccuracies in model parameterizations 66 (Carbone and Tuttle, 2008; Reif and Bluestein, 2017; Weckwerth et al., 2019; Zhao et 67 al., 2025). Crucially, the low-level jet (LLJ) that exhibit a diurnal cycle with a 68 maximum at night is widely recognized as a key contributor to nocturnal HR (Bonner 69 1968; Mitchell et al., 1995; Tuttle and Davis, 2006), as documented in regions or 70 countries such as the Great Plains of the United States (Maddox, 1983; Higgins et al., 71 1997), Argentina (Marengo et al., 2004), India (Monaghan et al., 2010), North China 72 Plain (Li et al., 2024). 73 The LLJs primarily originate from the inertial oscillations (IO) following the 74 sudden decay of turbulence after sunset (Blackadar, 1957) and thermal imbalances 75 induced baroclinicity over sloping terrain (Holton, 1967). Functioning as concentrated 76 corridors for heat, moisture, and momentum transport, LLJs can modulate the diurnal 77 oscillation in water vapor by IO (Rasmusson, 1967; Zhang et al., 2019) and enhance 78 convective instability, particularly when elevated high- $\theta_e$  air encounters frontal 79 boundaries (Trier et al., 2017). Also, strong low-level vertical wind shear (VWS) 80 associated with LLJs necessarily benefits deep lifting (Maddox et al., 1979; Stensrud, 81 1996; Rasmussen and Houze, 2016). These mechanisms collectively provide essential 82 thermodynamic and dynamic support for the initiation and organization of nocturnal 83 convection, especially where LLJs force low-level ascent at jet termini or via positive 84 vorticity advection left of the jet axis (Chen et al., 2017; Du and Chen, 2019; Xia and 85 Zhao, 2009). 86 Furthermore, LLJs interact synergistically with other key factors to trigger HR, 87 including terrain effects (Anthes et al., 1982; Pan and Chen, 2019; Huang et al., 2020), 88 gravity waves (Weckwerth & Wakimoto, 1992), and mesoscale systems (Chen et al. 89 2010; Chen et al., 2017), among others. These interactions are highly sensitive to the

91 Pu, 2019) and fine-scale structural of LLJs, including LLJ frequency, spatial 92 redistribution, and particularly localized wind profile accelerations (Pitchford and 93 London, 1962; Walters and Winkler, 2008; Du and Chen, 2019; Li et al., 2024). 94 Understanding these intricate evolution features of LLJs is critical for improving the 95 forecasting of nocturnal HR. 96 Despite advances facilitated by regional reanalysis (e.g., Doubler et al., 2015; Li 97 et al., 2021), numerical modeling (e.g., Zhang and Meng, 2019), radiosonde 98 observations (e.g., Whiteman et al., 1997; Yan et al., 2020), and emerging artificial 99 intelligence techniques (e.g., Subrahmanyam et al., 2024) in understanding the 100 climatology and physical mechanisms of LLJs and their role in HR forecasting, 101 significant knowledge gaps remain. However, a critical shortcoming lies in the inability 102 to capture minute-scale evolution of LLJs during the nocturnal pre-storm period. 103 Conventional observing systems lack the spatiotemporal resolution required to resolve 104 rapid changes in pre-storm environments (Weisman et al., 2015; Cao et at., 2025; Roots 105 et al., 2025), thereby hindering systematic analysis of fine-scale structure of LLJs and 106 their evolution within the critical 2-hour window preceding rainfall. 107 Moreover, the mechanisms and impacts of LLJs exhibit considerable variation 108 across monsoon phases and geographic regions. As a classic monsoon climate region, 109 China exhibits particularly prominent nocturnal rainfall contributions across major 110 climate-sensitive areas (Yu et al., 2014), where LLJs play a crucial role in modulating primary rainfall belts (Sun, 1986; Chen et al., 2010; Wang et al., 2013; Horinouchi et 111 112 al., 2019), such as those in Eastern China (Chen et al., 2017; Xue et al., 2018) and South 113 China (Du et al., 2020; Bai et al., 2021; Fu et al., 2021). However, nationwide 114 comparative studies examining LLJ precursor signals across different monsoon phases 115 in China are still lacking. 116 Radar wind profilers (RWPs) can offer transformative potential by capturing minute-resolution wind profiles to reveal pre-rainfall dynamic precursors (Zamora et 117 118 al., 1987; Du et al., 2012; Molod et al., 2019; Guo et al., 2023). For example, Gebauer

prevailing synoptic and subsynoptic-scale environmental conditions (e.g., Hodges and

119 et al. (2018) demonstrated the capability of RWPs to elucidate how heterogeneous 120 structures of LLJ trigger nocturnal convection in Great Plains; Based on a linear net of 121 RWPs deployed across the North China Plain, Li et al. (2024) observed rapid 122 intensification of MFC driven by a surge in LLJs profile within 30 min preceding 123 nocturnal rainfall onset, highlighting the sensitivity of RWP to minute-scale 124 perturbations of LLJs profiles. 125 Therefore, this study utilizes a nationwide network of RWPs to address the 126 following two questions: 1) How do the vertical structure of LLJs and their minute-127 scale evolution within 0-2 hours preceding nocturnal rainfall vary across different rainy

season phases? and 2) What are the systematic differences in LLJ dynamic-thermodynamic mechanisms between LLJ-influenced HR and non-HR events? The remainder of this paper is structured as follows: Section 2 details data and methodology, Section 3 presents comparative analyses of characteristics of rainfall and LLJs

Section 3 presents comparative analyses of characteristics of rainfall

evolution, and Section 4 synthesizes key conclusions.

# 134 **2. Data and Methodology**

128

129

130

143144

# 135 2.1 Radar wind profiler measurements

The RWP observations collected from 31 stations across China (Fig.1) from April to October in 2023-2024 were analyzed in this study, which can provide wind speed and direction with a vertical resolution of 120 m and an interval of 6 minutes (Liu et al., 2019). To reduce the potential influence of poor data quality, RWP data underwent strict quality control following procedures proposed by Wei et al. (2014) and Miao et al. (2018). Firstly, observations recorded during rainfall periods were eliminated. Secondly, for each profile, if more than 40% of the data points below 3 km above ground level (AGL) were outliers or missing, that entire profile was discarded. Next, within each profile below 3 km AGL, missing values and significant outliers that defined as values exceeding 2.5 standard deviations from the mean were removed. Finally, discontinuous, or missing data points were estimated using linear interpolation.

147 Following this quality control process, 109,400 wind profiles were discarded and a total 148 of 2,606,042 profiles across China were available for analysis during the study period. 149 2.2 Miscellaneous meteorological data 150 In addition, 1-min rainfall measurements were directly acquired from the rain 151 gauge measurements at 2160 national weather stations across China to identify rainfall 152 events. Rainfall amounts were accumulated over 6-min intervals to ensure temporal 153 alignment with the RWP measurements. Ground-based meteorological variables are 154 measured at 1-min intervals from national weather stations, including 2-m air temperature, relative humidity, and surface pressure. All ground-based data have 155 156 undergone rigorous quality control and are publicly available by the China 157 Meteorological Administration (CMA). 158 Furthermore, to diagnose large-scale circulation patterns and environmental 159 conditions preceding nocturnal rainfall influenced by LLJs, this study utilized 160 meteorological variables derived from the fifth generation of the European Centre for 161 Medium-Range Weather Forecasts atmospheric reanalysis (ERA5) of the global 162 climate (Hersbach et al., 2020). The ERA5 data features a horizontal resolution of 163 0.25°×0.25° across 37 vertical pressure levels and hourly temporal resolution. Unless 164 otherwise specified, all datasets cover the study period of April to October in 2023-165 2024. 166 2.3 Identification of nocturnal rainfall events 167 Firstly, days with typhoon activity were excluded. To minimize the impact of 168 rainfall on RWP measurements, a minimum dry interval of 2 hours was required 169 between consecutive rainfall events. Following the methodology of Li et al. (2024), a 170 rainfall occurrence was defined when the total accumulated rainfall which was 171 measured by all rain gauges within a 25-km radius of each RWP station exceeded 0.1 172 mm. Accounting for rainfall intermittency, a valid rainfall event required at least two 173 subsequent occurrences within 30 min following initial detection. Any isolated initial

occurrence not meeting this criterion was discarded.

Local Standard Time (LST). Based on operational classifications from the National 177 Water Resources Bureau and CMA, the rainy season was categorized into four 178 consecutive phases: (1) the South China Pre-summer Rainy Season (April 1 to June 8, 179 2023 and April 1 to June 9, 2024), (2) the Meiyu Season (June 9 to July 14, 2023 and 180 June 10 to July 21, 2024), (3) the North China Rainy Season (July 22 to August 31, 181 2023 and July 15 to August 31, 2024), and (4) the West China Autumn Rainy Season 182 (September 1 to October 31 for both 2023 and 2024), These phases are subsequently 183 designated as Phase 1 to Phase 4 throughout this study. Four regions of interest (ROIs) 184 were subsequently selected for detailed analysis (see Table 1). 185 Further screening identified nocturnal HR events where the mean 6-min rainfall 186 intensity exceeded the 75th percentile of all rainfall events at each station. Statistical 187 analysis revealed 3,155 nocturnal rainfall events during the 2023–2024 warm seasons (within the 31 red circles shown in Fig.1). Event counts per rainy season phase were 188 189 1,109, 689, 652, and 705 respectively, with 841 events classified as nocturnal HR 190 events. 191 2.4 Identification of LLJs and associated rainfall event 192 To ensure identified LLJs exhibit significant vertical wind shear characteristic of 193 jet-like profiles, the following criteria are adopted: (1) a maximum horizontal wind 194 speed exceeding 10 m s<sup>-1</sup> in the lowest 3 km AGL, and (2) a wind speed reduction of at 195 least 3 m s<sup>-1</sup> from the maximum to minimum below 3 km AGL, or to 3 km AGL if no 196 minimum exists (Bonner, 1968; Whiteman et al., 1997; Du et al., 2014; Yan et al., 2020). 197 These deliberately conservative wind speed thresholds maximize LLJ sample size for 198 enhanced statistical robustness. The strength of LLJ or jet nose is defined as the 199 maximum wind speed along the entire profile. The LLJ core height is defined as the 200 altitude of the wind speed maximum during LLJ occurrences. Correspondingly, the LLJ 201 direction is determined by the wind direction at the height of the LLJ. 202 We define rainfall events where LLJ occurs at least twice within 2 hours before 203 rainfall as an LLJ event. And HR events influenced by LLJs (LLJ HR events), HR

Nocturnal rainfall events were defined as those occurring between 2000 and 0800

events without LLJ influence (non-LLJ\_HR events), and non-HR events affected by

LLJ (LLJ non-HR events) are further distinguished.

227228

229

230

231

#### 3. Results and discussion

3.1 General characteristics of nocturnal rainfall and LLJs

Firstly, we characterized the spatiotemporal patterns of rainfall and LLJs observed nationwide during the 2023-2024 warm season. Nationally, nocturnal rainfall accounted for 51.6% of total warm-season rainfall (Fig. 2g), with pronounced concentrations over North, East, and Southwest China. In contrast, the pronounced daytime rainfall dominance in South China may arise from sea-breeze-front convective systems triggered by afternoon land-sea thermal contrasts and rapid destabilization of the local boundary layer. The frequency of nocturnal rainfall (52.5%) notably exceeded that in the daytime (47.5%) at the national scale, with the highest values occurring over southwestern and eastern regions (Fig. 2h). While the North China displayed distinct characteristics of low-frequency but high-intensity nocturnal rainfall. Although mean nocturnal rainfall intensity was generally lower, the probability of HR was higher at night than during the day, particularly across western China, North China, and northeastern China (Fig. 2i). Figure 3 displayed key attributes of LLJs detected at all 31 RWPs across China using Section 2 criteria. Compared to daytime, the occurrence frequency of nocturnal LLJs increased significantly by nearly 18%, with the most pronounced enhancement observed particularly over northern and eastern regions where nocturnal rainfall is more intense. Nocturnal LLJs exhibited stronger strength of core and highly occurrences below 1 km AGL, reflecting enhanced nocturnal LLJs within the boundary layer vertically. This vertical restructuring likely responds to nocturnal surface cooling and resultant low-level wind-field modifications. The dominant wind direction shifted to southerly or southwesterly flows at night across most regions, potentially driven by thermal contrasts within the monsoon circulation pattern and topographic forcing.

Statistical analysis revealed substantial linkage between LLJs and nocturnal 233 rainfall. Specifically, 56% of nocturnal rainfall events across China were preceded by 234 the presence of LLJs within 2 hours. Consequently, nocturnal rainfall influenced by 235 LLJs represents a major component of warm-season rainfall in China. And this 236 relationship was modulated by the seasonal migration of the western Pacific subtropical 237 high (WPSH), which drove corresponding shifts in HR belts that closely synchronized 238 with the spatiotemporal evolution of LLJ activity. The proportion of nocturnal rainfall 239 events associated with LLJs during the four rainy season phases reached 60.4%, 56.3%, 240 49.4%, and 54.9%, respectively (Fig. 4a). Among 841 identified nocturnal HR events, 241 the percentages classified as LLJ\_HR events were 47.2%, 43.6%, 33.9%, and 45.5% 242 from Phase 1 to Phase 4. For non-HR events, LLJ non-HR events accounted for 243 approximately 37.8%. These statistics collectively demonstrate the crucial role of LLJs 244 in initiating and modulating nocturnal rainfall across China. 245 Spatial analysis of rainfall intensity further revealed that LLJ\_HR events 246 consistently produced heavier rainfall than non-LLJ HR events, particularly within the 247 four ROIs identified in each phase (red boxes in Fig. 5). Therefore, a total of 142, 118, 248 63, and 59 nocturnal HR events were identified in these ROIs during from Phase 1 to 249 Phase 4 (Fig. 4b). From the LLJs perspective, nearly 31.1% of LLJ events were 250 classified as HR events across phases relative to LLJ non-HR events, which suggests 251 the presences of LLJs do not inherently increase the occurrence of HR. From the HR 252 perspective, nearly 45.0% of HR events were associated with LLJs within 2 hours 253 before occurred relative to non-LLJ HR events (see the pie charts in Fig. 6). And the 254 highest proportion of LLJ HR events occurred over ROI-2 during Phase 2, while the 255 lowest occurred in ROI-4 during Phase 4. In general, while nearly half of HR events 256 involved LLJs, merely 30% of LLJ events produced HR, indicating LLJs are necessary 257 but insufficient for HR. 258 Furthermore, probability distributions of rainfall intensity (Fig. 6) indicated that 259 LLJ HR events generally produced heavier rainfall at the national scale compared to 260 non-LLJ HR events, though the differences were less pronounced during Phases 2 and 261 4. Within the key regions, both event types showed higher probabilities of HR than the

262 national average, particularly for LLJ HR events. LLJ HR events in ROI-1 and ROI-263 2 demonstrated significantly higher probabilities of extreme rainfall (≥ 2 mm/6 min), 264 while those in ROI-4 during Phase 4 favored intensities near 0.5 and 2.8 mm/6 min. 265 Notably, although ROI-3 in Phase 3 exhibited a relatively high proportion of LLJ HR 266 events, their probability of producing heavier rainfall was lower than for non-LLJ HR 267 events, suggesting that LLJs may not represent the dominant mechanism for extreme 268 rainfall in this particular region and season. 269 3.2 Minute-scale evolution of LLJs preceding nocturnal heavy and non-heavy 270 rainfall 271 Although previous analyses have established the importance of LLJs in HR events, 272 it is noteworthy that approximately 70% of LLJ occurrences were not accompanied by 273 concurrent HR (i.e., LLJ non-HR events). To elucidate the contrasting precursor 274 characteristics of LLJs that lead to nocturnal rainfall of differing intensities, this section 275 examines fine-scale vertical structure and continuous evolution of LLJs within 2 hours 276 preceding both LLJ HR and LLJ non-HR events during four phases in their respective 277 ROIs. The results reveal distinct spatiotemporal variations in vertical structure and 278 evolutionary patterns of LLJs. 279 During Phase1 in ROI-1, LLJ HR events exhibited a significant increase in 280 frequency starting 108 min before rainfall onset, reaching secondary peaks at -84 min 281 and -60 min and culminating in maximum frequency immediately preceding rainfall 282 (Fig. 7a). These LLJs featured a bimodal vertical distribution with frequent occurrence 283 layers at 0.5-1 km and 1.5-2 km AGL. Wind profiles showed that LLJs strength 284 maximized at 48 min preceding rainfall accompanied by a slight downward extension, 285 followed by a modest weakening (Fig. 8a). In contrast, LLJ non-HR events exhibited 286 lower overall frequency predominantly below 1 km AGL, although they showed 287 gradual frequency increases from -48 min alongside strengthening winds and rising jet 288 heights (Figs. 7e and 8e). 289 However, both event types exhibited notably high frequencies and intensities of 290 LLJs over ROI-2 during Phase 2. For LLJ\_HR events, the total frequency and height

291 of LLJ occurrence initially declined followed by a rise. At -120 min, the frequency of 292 LLJs peaked and strength of jet profile reached a maximum (exceeding 12 m s<sup>-1</sup>). These 293 values then rapidly declined to a minimum at -84 min, concurrent with the jet height 294 descending from 1.5-2 km to below 1 km AGL. Then frequency of LLJs subsequently 295 increased with gradual height recovery, reaching a secondary peak at -48 min while 296 intensity of jet nose stabilized near 11 m s<sup>-1</sup>. In comparison, LLJ non-HR events 297 maintained consistent LLJs strength (near 11.8 m s<sup>-1</sup>) and a preferred height range of 298 1–2 km AGL (Fig. 8f). And LLJs frequency exhibited gradual changes, culminating in 299 a single peak 36 min before rainfall, followed by a rapid decrease (Fig. 7f). 300 During Phase 3 in ROI-3, LLJ\_HR events exhibited a bimodal temporal 301 distribution in LLJ frequency, with distinct peaks occurring at -96 min and -48 min 302 prior to rainfall (Fig. 7c). The dominant LLJ height was centered between 1-1.5 km 303 AGL. Beginning approximately 30 min before rainfall onset, a discernible descent of 304 the jet core was observed, accompanied by accelerated intensification of the wind 305 profile, which reached its maximum strength by -48 min (Fig. 8c). This rapid vertical 306 restructuring of LLJ is likely a key factor influencing HR occurrence. However, 307 markedly weaker wind profiles (near 9 m s<sup>-1</sup>) potentially explained the lower probability of heavier rainfall in LLJ HR events over ROI-3 during Phase 3 relative to 308 309 other region and phases. Conversely, LLJ non-HR events were characterized by lower 310 LLJs frequency and a more uniform height distribution (Fig. 7g). 311 During Phase 4 in ROI-4, LLJ HR events showed a rapid frequency increase from 312 48 min prior to rainfall. The LLJs were concentrated between 0.5–1.5 km AGL, and the 313 intensity of wind profile intensified progressively, with average maximum peaking near 314 13 m s<sup>-1</sup> just before rainfall onset (Fig. 8d). Conversely, LLJ non-HR events peaked 315 earlier at -84 min in both frequency and intensity of LLJs, followed by general 316 attenuation (Fig. 7h). By -48 min, wind profiles stabilized into a double-core structure 317 maintaining around 10 m s<sup>-1</sup>, with distinct jet cores near 0.8 km and 1.7 km AGL (Fig. 318 8h). 319 Furthermore, probability distributions of LLJ strength and height within 2 hours 320 preceding rainfall were compared across key regions (Fig. 9). During Phase 1 in ROI-

321 1, the strength of LLJs in LLJ HR events was notably stronger by 2-3 m s<sup>-1</sup> than that 322 in LLJ non-HR events (Fig. 9a). Height distributions showed distinct bimodal peaks 323 near 0.9 km and 1.75 km AGL (Fig. 9e), consistent with previously documented double 324 LLJs influences (Du and Chen, 2019). And average LLJs height was generally higher 325 in LLJ HR events. During Phase 2 in ROI-2, LLJ HR events showed higher 326 probabilities of strong LLJs (17–28 m s<sup>-1</sup>) compared to the dominant 13 m s<sup>-1</sup> intensity 327 in LLJ non-HR events (Fig. 9b). Influenced by large-scale circulation patterns, both 328 event types featured LLJs centered near 1.5 km AGL (Fig. 9f), though LLJ HR events 329 developed a secondary maximum near 0.8 km AGL due to pre-rainfall descent of the 330 jet core (Fig. 7b). Contrastingly, Figure 9c shows that LLJ\_HR events were associated 331 with weaker jet strengths (around 11 m s<sup>-1</sup>) compared to LLJ non-HR events (14-23 332 m s<sup>-1</sup>) in ROI-3 during Phase 3, suggesting that strong LLJs don't necessarily induce 333 HR here. The height of LLJ in LLJ HR events mainly concentrated near 1.2 km AGL, 334 whereas in LLJ non-HR events, it was more uniformly distributed between 0-3 km 335 AGL with a higher probability above 1.7 km (Fig. 9g). For Phase 4 in ROI-4, LLJs 336 strength peaked near 15 m s<sup>-1</sup> in both event types, but LLJ HR events featured stronger 337 jets reaching 25-30 m s<sup>-1</sup> (Fig. 9d). And LLJs height in both events peaked 338 predominantly at 0.8 km AGL, with secondary peaks at 1.5 km for LLJ HR and 2.0 km 339 AGL for LLJ non-HR events (Fig. 9h). 340 In summary, distinct temporal evolution patterns in frequency, occurrence height, 341 and wind profile intensity of LLJs preceding LLJ HR events were observed across key 342 regions during each phase. Except for ROI-3 during Phase 3, LLJ HR events exhibited 343 significantly higher probabilities of stronger jets, most notably in ROI-2 during Phase2. 344 LLJs height were generally lower in LLJ HR events, except in ROI-1 during Phase1. 345 These findings highlight the role of fine-scale LLJ structures and their rapid vertical 346 reorganization in modulating nocturnal rainfall intensity, offering valuable insights for 347 improving regional nocturnal HR forecasting.

348 3.3 Thermodynamic evolution associated with LLJs preceding nocturnal heavy 349 and non-heavy Rainfall 350 The preceding section 3.2 has clarified that the fine-scale dynamic characteristics 351 of LLJs-including their temporal evolution, vertical structure, and intensity 352 variations—play a pivotal role in modulating nocturnal rainfall intensity across 353 different key regions and rainy season phases. However, the influence of LLJs on 354 rainfall generation and intensification rarely operates in isolation; instead, it depends 355 strongly on the accompanying large-scale thermodynamic environment, which 356 provides the necessary moisture supply and convective instability to sustain or amplify 357 heavy rainfall. Thus, to fully unravel the mechanisms underlying the distinction 358 between LLJ HR and LLJ non-HR events, it is essential to complement the dynamic 359 analysis with an in-depth examination of the thermodynamic conditions associated with 360 LLJs in the 1-hour preceding nocturnal rainfall. 361 Further analysis of large-scale thermodynamic conditions at 1-hour preceding 362 rainfall (Figs. 10 and 11) reveals consistently stronger thermal instability for LLJ HR 363 versus LLJ\_non-HR events, accompanied by stronger MFC within key regions during 364 each rainy season. 365 During Phase 1 in ROI-1, thermodynamic conditions were comparable between 366 event types. Southwesterly LLJs transported warm-moist air masses from the South 367 China Sea and Bay of Bengal, forming a pronounced warm-humid tongue. Coupled 368 with MFC centers developing north of the jet axis, this configuration facilitated nocturnal rainfall development. During Phase 2 in ROI-2, although warm-moist airflow 369 370 prevailed northwest of the subtropical high in both event types, LLJ HR events 371 exhibited more favorable thermodynamic conditions: mean  $\theta_e$  was approximately 2 K 372 higher than in LLJ non-HR events, with a high- $\theta_e$  center reaching 358 K, alongside stronger LLJ cores (>1.2 m s<sup>-1</sup> difference). Through baroclinic instability and LLJ 373 374 dynamic forcing, strong MFC centers formed along the left side of the jet axis, 375 ultimately triggering HR. During Phase 3 in ROI-3, LLJ HR events were characterized 376 by enhanced southwesterly moisture transport interacting with cold air advection from

378

379380

398399

westerly troughs. Superimposed with topographic forcing from the Taihang Mountains, this resulted in MFC intensities south of Beijing nearly  $30 \times 10^{-5}$  km m<sup>-2</sup>s<sup>-1</sup> than those in LLJ non-HR events, thereby driving nocturnal HR. Thermodynamic contrasts were most pronounced during Phase 4 in ROI-4. Unlike LLJ non-HR events characterized by southwestward-extending cold highs, LLJ HR events developed deep high- $\theta_e$ centers (>360 K) over the southeastern Tibetan Plateau, supplying abundant moisture and convective instability. Concurrently, stronger anticyclonic circulation (consistent with enhanced LLJ winds in Fig. 8d) drove cold air intrusion into ROI-4. This cold airflow converged with warm-moist plateau currents over the Sichuan Basin, synergizing with orographic forcing to generate a strong MFC center on the eastern lee slope of plateau to promote nocturnal HR in ROI-4. To elucidate the rapid processes leading to occurrence of rainfall, the minute-scale evolution of key thermodynamic and dynamic parameters was further analyzed (Fig. 12), including surface  $\theta_e$ , LLJ index (defined as the ratio of maximum wind speed below 3 km to the height where wind first exceeds 10 m s<sup>-1</sup>) and VWS (calculated as the wind speed difference between the surface and jet height divided by the jet height). Figure 12a shows that during Phase 1 in ROI-1, although large-scale environments were similar between event types (LLJ HR and LLJ non-HR), the rapid, minute-scale co-intensification of thermodynamic and dynamic process prior to rainfall onset governs the resulting rainfall intensity. LLJ\_HR events exhibited abrupt thermodynamic enhancement at 90 min preceding the onset of rainfall, with surface  $\theta_e$ and VWS surging approximately 1.5 K and 0.5 s<sup>-1</sup> respectively. Concurrently, the LLJ index surged by 0.03 and VWS peaked sharply at -60 min, signaling LLJs intensification and core descent (Figs. 7a, 8a). This coordinated intensification suggests a tightly coupled process in which low-level warming and moistening reinforced wind acceleration, thereby enhancing shear and promoting deeper lifting conducive to HR. In contrast, LLJ non-HR events showed weaker increases and a declining LLJ index (by about 0.015) alongside rising jet cores, reducing low-level shear and convergence efficiency, thereby diminishing overall rainfall intensity.

During Phase 2 in ROI-2, LLJ HR events showed concurrent peaks in dynamic 407 parameters (LLJ index, strength and VWS) coupled with high  $\theta_e$  (348.3K) at -120 min 408 (Fig. 12b). At -84 min, a critical rapid descent of jet core height below 1 km AGL 409 coincided with surface warming and a LLJ index maximum. This change likely 410 enhanced boundary-layer convergence by bringing stronger winds into closer proximity 411 to the surface, effectively steepening VWS and facilitating deeper convective initiation. 412 The timing of this rapid evolution suggests a transient optimal window for nocturnal 413 rainfall triggering that is characteristic of LLJ HR events in ROI-2. Subsequently, the 414 LLJ index declined sharply and a rapid surface cooling initiates beginning 60 min prior 415 to HR likely resulted from the outflow of cold pool outflow associated with alternation 416 of convective systems in the Mei-Yu front cloud system (Zhang et al., 2023). The 417 combination of the cold air and the strong southwestly LLJs may have further enhanced 418 uplift and promoted rainfall (Luo et al., 2014). For LLJ non-HR events, weaker 419 thermodynamic support and diminished dynamic forcing with consistently lower LLJ 420 indices (∆I≈0.015) within 60 min preceding rainfall resulted in insufficient lift to 421 sustain HR. These contrasting features underscored the importance of thermal-dynamic 422 synergy influenced by LLJs evolution in triggering HR. 423 During Phase 3 in ROI-3, LLJ HR events featured prominent thermal 424 compensation (surface  $\Delta \theta_e > 1$ K, 850hPa  $\Delta \theta_e > 2$ K versus non-HR events) despite 425 weaker dynamics compared to other phases (Fig. 12c). The LLJ index progressively 426 increased while VWS peaked synchronously with LLJ frequency. Within 30 min 427 preceding rainfall, substantial intensification of VWS by about 1.5 m s<sup>-1</sup> occurred 428 driven by a rapid acceleration in the LLJ wind profile. This co-evolution with rapid 429 surface warming (increase of 0.25 K) released convective instability, enhancing 430 boundary-layer convergence (Fig. 10) and explaining the stronger MFC observed. 431 Nevertheless, the overall weaker dynamical conditions likely limited the depth and 432 organization of convection, explaining the reduced probability of heavier rainfall 433 compared to other phases. 434 And LLJ HR events showed explosive dynamic intensification from 48 min preceding rainfall in ROI-4 during Phase 4. with VWS surging 0.9 s<sup>-1</sup> and LLJ index 435

0.025 (Fig. 12d). This rapid strengthening of the jet occurred concurrently with a gradual rise in  $\theta_e$  (>346 K), enhancing low-level moist static energy and supporting vigorous uplift along the windward slopes of the Sichuan Basin. The synergistic coupling between dynamic forcing and thermodynamic persistence underscores the critical role of minute-scale jet intensification as a precursor to nocturnal HR under favorably pre-conditioned environments in ROI-4. For LLJ\_non-HR events, VWS and surface  $\theta_e$  reached their peaks almost simultaneously at 84 min before rainfall followed by gradual decline. The premature peak and subsequent decay in both dynamic and thermodynamic fields resulted in a lack of sustained forcing during the critical pre-convective period. And these events exhibited weaker changes of dynamics with marginal temporal evolution (VWS 

466 the strong linkage between LLJs and nocturnal HR. 467 Pronounced differences and varying lead times were observed in vertically 468 resolved evolution of LLJs and their thermodynamic environments when comparing LLJ HR and LLJ non-HR events. During Phase 1 in ROI-1, LLJ HR events were 469 470 marked by a sharp rise in LLJ frequency beginning 108 min before rainfall, accompanied by abrupt thermodynamic intensification ( $\Delta\theta_e \approx 1.5$ K,  $\Delta VWS \approx 0.5 \text{ s}^{-1}$ ) 471 from -90 min. In contrast, LLJ non-HR events exhibited weaker and more stable 472 thermo-dynamic conditions. During Phase 2 in ROI-2, LLJ HR events were preceded 473 474 by initially strong jets (>12 m s<sup>-1</sup>) and elevated surface  $\theta_e$  (~348.3 K), followed by a 475 rapid descent of the LLJ core below 1 km AGL at -84 min, leading to a peak in the LLJ 476 index. While LLJ non-HR events were characterized by a stable jet core but declining 477 dynamic indices and frequency within the final 60 min before rainfall. During Phase 3 478 in ROI-3, LLJ HR events exhibited higher LLJ frequency and an increasing LLJ index, 479 culminating in rapid wind profile restructuring within 30 min before rainfall, all under 480 significantly more unstable thermal conditions (surface  $\Delta \theta_e > 1$  K) compared to non-481 HR events. During Phase 4 in ROI-4, LLJ HR events were driven by favorable 482 thermodynamic environments and minute-scale dynamical changes. A surge in jet frequency at -48 min triggered abrupt increases in VWS (≈0.9 s<sup>-1</sup>) and the LLJ index 483 484 ( $\approx$ 0.025) accompanied with high  $\theta_e$  air, enhancing strong MFC over the southeastern 485 plateau. By contrast, LLJ non-HR events lacked adequate thermal forcing, and LLJ 486 intensity, frequency, and dynamical parameters showed a consistent declining trend. 487 Crucially, A key finding is that nocturnal LLJ HR events universally require rapid, 488 coupled minute-scale dynamical intensification occurring 30-120 min preceding 489 rainfall onset, acting in synergy with thermodynamic instability. LLJ non-HR events 490 consistently displayed insufficient dynamic-thermodynamic coupling, reflected in 491 stable or declining LLJ parameters. This underscores that the intensity and timing of 492 nocturnal rainfall are ultimately regulated by regionally specific thermo-dynamic 493 interactions modulated by the evolution of fine vertical structure of the LLJ.

higher probability of producing heavier rainfall than non-LLJ HR events, underscoring

This study establishes distinct dynamic-rainfall linkages associated with LLJs across different warm-season rainy periods in China. Future research should: (1) expand multi-source observations to establish dynamic thresholds for early forecasting systems of nocturnal rainfall, and (2) develop quantitative frameworks relating LLJ structural evolution to rainfall intensity, offering theoretical support for optimizing physical processes in LLJ parameterization schemes within high-resolution numerical models. Additionally, the physical mechanisms governing evolution of LLJs height or strength immediately preceding rainfall onset require further investigation. **Data Availability** The LLJs retrieved from the RWP network can be acquired from https://doi.org/10.5281/zenodo.17176759 (Li and Guo, 2025). The data from the weather station are obtained from the China Meteorological Data Service Centre at https://data.cma.cn/en, and the original ERA5 reanalysis data used here are available from the ECMWF in Hersbach et al. (2020). Acknowledgments This work was supported by the National Natural Science Foundation of China under Grants of 42325501 and 42105090, and by the National Key Research and Development Program of China under grant 2024YFC3013001. Last but not least, we appreciated tremendously the constructive comments and suggestions made by the anonymous reviewers that significantly improved the quality of our manuscript. **Author Contributions** The study was completed with close cooperation between all authors. JG designed the research framework; NL performed the analysis and drafted the original manuscript with contribution from JG; JG, XG, ZZ, YZ. JG, NT and YW helped revise the manuscript.

**Completing interests** 

The authors declare that they have no conflict of interest.

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

| 727 | Zhang, Y., Xue, M., Zhu, K., and Zhou, B.: What is the main cause of diurnal variation |
|-----|----------------------------------------------------------------------------------------|
| 728 | and nocturnal peak of summer precipitation in Sichuan Basin, China? the key role       |
| 729 | of boundary layer low-level jet inertial oscillations, J. Geophys. Res. Atmos.,        |
| 730 | 124(5), 2643-2664, doi:10.1029/2018jd029834, 2019.                                     |
| 731 |                                                                                        |
| 732 |                                                                                        |

**Table 1.** Table of Representative Radar Wind Profiler Stations in Mainland China

| Region   | Station |                  | Longitude (°) | Latitude (°) | Altitude (m) |
|----------|---------|------------------|---------------|--------------|--------------|
|          | 58839   | MQ               | 118.86        | 26.22        | 160.70       |
|          | 59046   | LZ               | 109.46        | 24.36        | 314.40       |
| ROI-1    | 59137   | JJ               | 118.54        | 24.81        | 124.80       |
|          | 59287   | GZ               | 113.48        | 23.21        | 65.00        |
|          | 59758   | HK               | 110.25        | 19.99        | 69.00        |
|          | 57461   | YC               | 111.36        | 30.74        | 253.80       |
|          | 57687   | CS               | 112.79        | 28.11        | 119.00       |
|          | 57793   | YCN              | 114.36        | 27.79        | 132.00       |
| ROI-2    | 58238   | $_{\mathrm{BJ}}$ | 118.90        | 31.93        | 40.60        |
|          | 58321   | HF               | 117.03        | 31.57        | 50.00        |
|          | 58367   | LH               | 121.47        | 31.18        | 5.00         |
|          | 58459   | XS               | 120.29        | 30.18        | 48.80        |
|          | 53463   | НННТ             | 111.68        | 40.82        | 1152.10      |
| DOL 2    | 53772   | TY               | 112.58        | 37.62        | 785.00       |
| ROI-3    | 54511   | $_{\mathrm{BJ}}$ | 116.47        | 39.81        | 31.50        |
|          | 54534   | TS               | 118.10        | 39.65        | 23.20        |
|          | 54602   | BD               | 115.48        | 38.73        | 16.80        |
|          | 57816   | GY               | 106.73        | 26.59        | 1197.60      |
| ROI-4    | 56290   | XD               | 104.18        | 30.77        | 514.00       |
|          | 56651   | LJ               | 100.22        | 26.85        | 2382.40      |
|          | 50936   | BC               | 122.47        | 45.36        | 156.00       |
|          | 51463   | WLMQ             | 87.65         | 43.79        | 935.00       |
|          | 51628   | AKS              | 80.38         | 41.12        | 1107.10      |
|          | 52754   | GC               | 100.08        | 37.2         | 3301.50      |
| Other    | 52889   | LZ               | 103.89        | 36.06        | 1519.20      |
| stations | 57516   | CQ               | 106.46        | 29.57        | 260.00       |
|          | 53845   | YA               | 109.45        | 36.58        | 1180.40      |
|          | 54342   | SY               | 123.51        | 41.73        | 50.00        |
|          | 54857   | QD               | 120.13        | 36.23        | 12.00        |
|          | 55664   | DR               | 87.07         | 28.63        | 4302.00      |
|          | 57171   | PDS              | 113.12        | 33.77        | 142.00       |

# 734 Figures

**Figure.1** (a) Spatial distribution of 31 Radar Wind Profiler (RWP) stations across China, with four regions of interest (ROIs) demarcated by blue dashed boxes: ROI-1, ROI-2, ROI-3, and ROI-3. (b) Schematic of spatial co-location: Beijing Observatory's RWP (red star) and rain gauges (black dots) within a 25-km radius

**Figure 2.** (a–c) Spatial distributions of accumulated rainfall (mm), rainfall frequency (%) and rainfall intensity (mm/h) in the daytime from April to October in 2023–2024. The numbers in the upper left corner represent the national average; (d–f) the same as (a–c), but in the nighttime; (g–i) Nocturnal contribution ratios of accumulated rainfall, frequency, and occurrence frequency of heavy rainfall (>75th percentile intensity). The pie charts illustrate the contribution rates of daytime (blue) and nighttime (red) at the national scale

**Figure 3.** (a–d) Spatial distribution of occurrence frequency, height, strength, and the dominant wind direction of LLJs observed by 31 RWP stations during April-October from 2023 to 2024 in the daytime. (e–h) Same as (a–d), but in the nighttime

**Figure 4.** (a) Statistic of all nocturnal rainfall events (solid-filled bars) and heavy rainfall (HR; diagonally striped bars) events selected across China during four phases, including LLJ events (red) and non-LLJ events (blue); (b) Same as panel (a), but for nocturnal HR events in from ROI-1 to ROI-4 during corresponding phase, including LLJ-HR (red), LLJ\_non-HR (yellow), non-LLJ\_HR (dark blue), and non-LLJ\_non-HR (light blue) events.

**Figure 5.** (a–d) Spatial distributions of average rain rate (mm/6 min) for nocturnal LLJ\_HR events during the warm season from Phase 1 to Phase 4 across China; (e–h) Same as (a–d), but for non-LLJ\_HR events. The red frame indicates four ROIs.

784

785

786

**Figure 6.** (a) Probability density distributions of average rain rate (mm/6 min) for LLJ-HR events (black solid lines) and non-LLJ\_HR events (gray solid lines) across China during Phase 1, and specifically in ROI-1 for LLJ-HR events (red solid lines) and non-LLJ\_HR events (blue solid lines). (b-d) the same as panel (a), but for comparisons between national-scale and other regional-scale events in ROI-2 during Phase 2, ROI-3 during Phase 3, and ROI-4 during Phase 4. The pie chart at the lower right shows the proportion distribution of LLJ-HR and non-LLJ\_HR events in these key regions during each period

**Figure 7.** Time-height evolution of LLJ occurrence frequency (color shading, every 12 min, within 500 m vertical bins) detected by RWP with 2 hours preceding nocturnal rainfall in LLJ-HR events in (a) ROI-1 during Phase 1, (b) ROI-2 during Phase 2, (c) ROI-3 during Phase 3, and (d) in ROI-4 during Phase 4. Black solid lines denote accumulated LLJ frequency for 3-km altitude. (e-h) Same as (a-d), but for LLJ\_non-HR events

**Figure 8.** (a-d) Evolution of RWP-detected mean wind profiles of LLJs (blue solid lines, every 12 min) within 2 hours preceding nocturnal rainfall in LLJ-HR events in (a) ROI-1 during Phase 1, (b) ROI-2 during Phase 2, (c) ROI-3 during Phase 3, and (d) in ROI-4 during Phase 4. (e-h) Same as (a-d), but for LLJ\_non-HR events

**Figure 9.** Probability density distributions of jet core intensity from RWP observations within 2 hours preceding nocturnal rainfall in LLJ-HR events in (a) ROI-1 during Phase 1, (b) ROI-2 during Phase 2, (c) ROI-3 during Phase 3, and (d) in ROI-4 during Phase 4. (e-h) Same as (a-d), but for the height of LLJs

**Figure 10.** Distributions of equivalent potential temperature (shading, unit: K) at 850 hPa, superimposed with 850 hPa horizontal wind vectors (black arrows) and geopotential height contours (red solid lines), for LLJ-HR events within 1-hour time window preceding nocturnal rainfall onset in (a) ROI-1 during Phase 1, (b) ROI-2 during Phase 2, (c) ROI-3 during Phase 3, and (d) in ROI-4 during Phase 4. Gray shading denotes terrain elevation exceeding 850 Pa level. The reference vector (10 m s<sup>-1</sup>) is shown at the lower-left corner. (e-h) Same as (a-d), but for LLJ\_non-HR events

**Figure 11.** Same as Figure 10, but showing the integrated moisture flux divergence (shading, unit: kg·m<sup>-2</sup> s<sup>-1</sup>) between 1000–700 hPa at 1 hour preceding nocturnal rainfall onset

829

830

831

Figure 12. Temporal evolution of surface equivalent potential temperature ( $\theta_e$ , red lines), vertical wind shear (VWS, blue lines), and LLJ index (I, black lines) averaged within 2 hours preceding nocturnal rainfall for LLJ-HR events (solid lines) and LLJ\_non-HR events (dashed lines) in (a) ROI-1 during Phase 1, (b) ROI-2 during Phase 2, (c) ROI-3 during Phase 3, and (d) in ROI-4 during Phase 4. Green bars denote 6-min averaged rain rate (mm/ 6 min) after LLJ-HR (solid bars) and LLJ\_non-HR (open bars) events onset