# Peer review of "Abstract 23 Nocturnal rainfall initiation is closely linked to low-level jets (LLJs), but national-scale 24 LLJ features over China—especially the"

_EGUsphere, 2025_

## Referee Comment (RC1)

Review of "On the Nationwide Variability of Low-Level Jets Prior to Warm-season Nocturnal Rainfall in China Revealed by Radar Wind Profilers" (author: Ning Li, Jianping Guo, Xiaoran Guo, Tianmeng Chen, Zhen Zhang, Na Tang, Yifei Wang, Honglong Yang, and Yongguang Zheng)

**Overview:**

RWPs are advanced equipment capable of capturing wind profile evolution on minute scale. This equipment offers valuable information and potential precursors for nowcasting rainfall and convection occurrence. It is particularly crucial for region like China that has established nationwide RWP network. For this purpose, this work employed 31 RWP stations across China, and explored their minute-scale information before nocturnal rainfall production, with emphasis on the indication by RWP-detected LLJ traits. Unique dynamic features leading to rainfall were revealed across distinct regions and phases across China. The differences in LLJ-involved heavy rainfall and non-heavy rainfall events were further examined from both dynamic and thermodynamic viewpoints. Overall, the findings are meaningful for advancing our knowledge on the LLJ-related activities and their interaction with other factors preceding rainfall production in different regimes. The manuscript is generally well-written. In my opinion, this paper is acceptable after some minor revisions.

**Minor comments:**

- 1. Abstract: The "multiscale responses" is ambiguous here.
- 2. Throughout the manuscript: Several sentences in the manuscript contain grammatical mistakes, such as those beginning with "While" and "And" (L217–218, L235–238, L253–255, L298–299, and elsewhere). There is also misuse of hyphen (-) and long dash (-), especially those between the months (April–October), years (2023–2024).
- 3. L50: Suggest to revise "The heavy rains linked to LLJs" as "The LLJs-associated heavy rain".
- 4. L88: Can you provide some examples for "mesoscale systems"? On the other hand, this term seems not parallel with "terrain effects" and "gravity waves" logically.
- 5. L149: Suggest to revise "Miscellaneous" as "Multi-source".
- 6. L156: Can you provide some reference for the quality control procedure of ground-based data?
- 7. L182: Revise symbol: comma to period.
- 8. L185–190: I am not so sure about the rationality and accuracy of HR definition here, given HR has its common-used definition in routine operation.
- 9. L194: Add space before 10 m s-1. The authors are encouraged to provide reference for the choice of this LLJ magnitude.
- 10. L203: should be "an LLJ-associated rainfall event"?
- 11. L211: The contribution rate is "50.9%" in Fig. 2g, rather than "51.6%".
- 12. L212–215: Please provide literature for this speculation.

- 13. L215–216: "52.5%" and "47.5%" is the relative contribution rate of frequency, but not the frequency of nocturnal rainfall or daytime rainfall. Please clarify the frequency at national scale that is quite confusing in current statement.
- 14. L222: "displayed" -> "displays".
- 15. L223–226: I think the more intense nocturnal rainfall should be attributed to the absolute magnitude of nocturnal LLJs. It seems unreasonable to relate this intense rainfall with the enhancement of nocturnal LLJs relative to daytime LLJs.
- 16. Figure 2: It seems (a)–(f) here are unnecessary to be shown since they are not cited in the text. It would be better to add relevant description for these subplots, or delete them.
- 17. Figure 3d-h: Why only four directions for LLJs, but not eight directions?
- 18. L247: Please consider deleting "Therefore". There is no causal relationship between the two sentences.
- 19. L251–253: The statement of "nearly 45.0%" cannot be well supported by the pie charts in Fig. 6.
- 20. Figure 7: Do the blue line denote the accumulated LLJ frequency over 0–3 km latitude?
- 21. L289: Consider deleting "However".
- 22. Figure 8c: The maximum wind speeds are lower than 10 m/s. I wonder how they satisfy the LLJ definition in section 2.
- 23. L322–324: The double LLJs in Du and Chen (2019) are at nearly 950 hPa (BLJ) and 700 hPa (SLLJ). How the bimodal peaks here correspond to their double LLJs?
- 24. L335: It should be "under 1.7 km"?
- 25. L360: The choice of 1 hour instead of 2 hours used before should be further explained.
- 26. L363: "MFC" stands for?
- 27. L365–387: The authors are encouraged to add more specific citation of Fig. 10 and Fig. 11 for these statements, for the convenience of reader.
- 28. L390–391: The physical meanings of "LLJ index" and "VWS" here are unclear, especially "LLJ index". Please try to clarify these terms.
- 29. L403: weaker increases in surface  $\theta_e$ ?
- 30. L279–318, L365–447: The presentation of the results across different regimes appear to be burdensome for readers. Please consider reorganizing these texts to highlight the similarities and differences among the regimes.

---

## Author Comment (AC1)

**Response to Reviewer #1' Comments**

**RWPs are advanced equipment capable of capturing wind profile evolution on minute scale. This equipment offers valuable information and potential precursors for nowcasting rainfall and convection occurrence. It is particularly crucial for region like China that has established nationwide RWP network. For this purpose, this work employed 31 RWP stations across China, and explored their minute-scale information before nocturnal rainfall production, with emphasis on the indication by RWP-detected LLJ traits. Unique dynamic features leading to rainfall were revealed across distinct regions and phases across China. The differences in LLJ-involved heavy rainfall and non-heavy rainfall events were further examined from both dynamic and thermodynamic viewpoints. Overall, the findings are meaningful for advancing our knowledge on the LLJ-related activities and their interaction with other factors preceding rainfall production in different regimes. The manuscript is generally well-written. In my opinion, this paper is acceptable after some minor revisions.**

Response: First of all, we thank you for taking the time to review our manuscript and offering constructive and thoughtful suggestions! Per your kind comments, we have carefully revised the manuscript. For clarity purpose, here we have listed the reviewers' comments in bold font, followed by our response in blue plain font, and the modifications to the manuscript are in blue italics.

**Minor comments:**

**1. Abstract: The "multiscale responses" is ambiguous here.**

Response: We agree that intended meaning of "multiscale responses"—the synthesis of data from multiple sources (RWP data, surface automatic weather station and reanalysis data) and multi-dimensions (from large-scale circulation to fine-scale vertical structure)—was not sufficiently clear and could lead to ambiguity.

To address this, we have revised the relevant sentence in the abstract to directly and precisely describe our methodological approach, which is shown as follows:

*"Here, we reveal the fine vertical structure of LLJs and their rapid evolution within 2 hours preceding the onset of nocturnal heavy rain (HR) and non-HR across four phases of rainy seasons in China during the warm season (April–October) of 2023–2024, utilizing data from a nationwide network of radar wind profilers (RWPs) in combination with surface observations and reanalysis data."*

**2. Throughout the manuscript: Several sentences in the manuscript contain grammatical mistakes, such as those beginning with "While" and "And" (L217–218, L235–238, L253–255, L298–299, and elsewhere). There is also misuse of hyphen (-) and long dash (–), especially those between the months (April– October), years (2023–2024).**

Response: Amended as suggested.

**3. L50: Suggest to revise "The heavy rains linked to LLJs" as "The LLJs-associated heavy rain".**

Response: Amended as suggested.

**4. L88: Can you provide some examples for "mesoscale systems"? On the other hand, this term seems not parallel with "terrain effects" and "gravity waves" logically.**

Response: This sentence has been rephrased as:

*"Furthermore, LLJs interact synergistically with other key factors to trigger HR that is associated with mesoscale convective systems (Chen et al. 2010; Chen et al., 2017; Chen et al., 2024), including terrain effects (Anthes et al., 1982; Pan and Chen, 2019; Huang et al., 2020), gravity waves (Weckwerth & Wakimoto, 1992), among others."*

**5. L149: Suggest to revise "Miscellaneous" as "Multi-source".**

Response: Amended as suggested.

**6. L156: Can you provide some reference for the quality control procedure of ground-based data?**

Response: Yes, we have added some reference for the quality control procedure of ground-based data in this revised manuscript:

*"All ground-based data have undergone rigorous quality control (China Meteorological Administration, 2020; Zhao et al., 2024) and are publicly available by the China Meteorological Administration (CMA)."*

**References**

Zhao, Y., Liao, J., Zhang, Q., Chen, J., Gong, X., Shi, Y., Shi, M., Yang, D., Fan, S., Zhou, X., Cao, L., and Hu, K.: Development of China Ground Climate Normal Value Dataset from 1991 to 2020, Chinese Journal of Atmospheric Sciences (in Chinese), 48(2), 555–571, doi:10.3878/j.issn.1006-9895.2204.22010, 2024.

China Meteorological Administration: Specification for Automatic Observation of Ground Meteorology. China Meteorological Press, 2020.

**7. L182: Revise symbol: comma to period.**

Response: Amended as suggested.

**8. L185–190: I am not so sure about the rationality and accuracy of HR definition here, given HR has its common-used definition in routine operation.**

Response: We think that utilizing the station-specific 75th percentile as the threshold is reasonable and necessary for this study, based on the following three considerations:

(1) Warm-season rainfall across China exhibits strong spatial heterogeneity. At present, in China's meteorological operations, the terms "$\geq 20$ mm/h" and "$\geq 50$ mm/day" are commonly used to define hourly heavy rainfall and rainstorm days respectively. Based on the definition method using a fixed threshold, it can reflect the absolute intensity of extreme precipitation but often miss locally significant events in arid regions and causes statistics bias of nationwide LLJ-associated rainfall events. Compared with business applications, the relative threshold method is more commonly

used to define heavy rain (Zhai et al., 2005; Zhang and Zhai, 2011; Xiao et al., 2016; Wu et al., 2019). By using a station-specific percentile threshold, we ensure the identification of locally significant rainfall events relative to the local climatology and can better compare local difference of fine-scale dynamical processes of LLJ before HR events versus non-HR events.

(2) The 75th percentile was chosen as the optimal threshold to effectively distinguish significant heavy rainfall from weak rainfall while ensuring a sufficient sample size for robust statistical analysis. As evidenced in Table S1, utilizing a stricter threshold (e.g., the 85th and 95th percentile) would drastically reduce the sample size to single digits in certain phases, rendering statistical conclusions unreliable and without statistical significance. The 75th percentile effectively retain a sufficient sample size to ensure statistical power while still effectively filtering out weak precipitation.

(3) Sensitivity analysis by varying the thresholds to 85th and 95th percentile to ensure our key conclusions are not artifacts of this specific threshold. As shown in Figure S1–S4, the results showed that although the specific lead times varied when changing the thresholds, LLJ_HR events still undergo a minute-scale "rapid reorganization" characterized by oscillations in jet height, frequency and strength, and the "final-stage intensification" of LLJs structure. LLJ_non-HR events still exhibit quasi-steady or declining dynamical response of LLJs. The main conclusion regarding the precursory signals of LLJs is robust within a reasonable threshold range.

Therefore, the 75th percentile serves as an optimal balance between maintaining statistical power and successfully distinguishing the essential difference of dynamical mechanisms prior to HR and non-HR events. Thank you for raising this important point regarding the percentile-based threshold for defining HR events, which prompted a thorough investigation. We have incorporated these justifications, and the sensitivity analysis results into Section 2.3 and the Supplementary Material of the revised manuscript.

Table S1 Statistics of the number of LLJ_HR and LLJ_non-HR events during four

rainy season phases under different percentile levels of rainfall intensity

| Type | Percentile | Phase 1 | Phase 2 | Phase 3 | Phase 4 |
|---|---|---|---|---|---|
| LLJ_HR | 75th | 29 | 26 | 15 | 12 |
| | 85th | 20 | 15 | 10 | 8 |
| | 95th | 9 | 8 | 5 | 4 |
| LLJ_non-HR | 75th | 71 | 69 | 30 | 25 |
| | 85th | 95 | 103 | 47 | 33 |
| | 95th | 110 | 112 | 53 | 38 |

[Figure]

**Figure S1.** (a-d) Evolution of RWP-detected mean wind profiles of LLJs (blue solid lines, every 12 min) within 2 hours preceding nocturnal rainfall in LLJ-HR events (≥85th percentile) in (a) ROI-1 during Phase 1, (b) ROI-2 during Phase 2, (c) ROI-3 during Phase 3, and (d) in ROI-4 during Phase 4. (e-h) Same as (a-d), but for LLJ_non-HR events

[Figure]

**Figure S2.** Same as Figure S1, but for HR events (≥95th percentile).

[Figure]

**Figure S3.** Time-height evolution of LLJ occurrence frequency (color shading, every 12 min, within 500 m vertical bins) detected by RWP with 2 hours preceding nocturnal rainfall in LLJ-HR (≥85th percentile) events in (a) ROI-1 during Phase 1, (b) ROI-2 during Phase 2, (c) ROI-3 during Phase 3, and (d) in ROI-4 during Phase 4. Dark blue solid lines denote accumulated LLJ frequency over 0–3 km latitude. (e-h) Same as (a-d), but for LLJ_non-HR events

[Figure]

**Figure S4.** Same as Figure S3, but for HR events (≥95th percentile).

**9. L194: Add space before 10 m s-1. The authors are encouraged to provide reference for the choice of this LLJ magnitude.**

Response: Amended as suggested.

**10. L203: should be "an LLJ-associated rainfall event"?**

Response: Actually, in our study, the term "LLJ event" specifically denote a category of rainfall events that are associated with LLJ activity. This definition is adopted for

brevity and is consistent with the convention established in prior literature in our field (e.g., Li and Du, 2021; Li et al., 2024). Therefore, this definition is clearly stated and believe it is sufficiently clear to the reader. Therefore, we have retained the original term "LLJ event" in our revised manuscript.

**11. L211: The contribution rate is "50.9%" in Fig. 2g, rather than "51.6%".**

Response: Amended as suggested.

**12. L212–215: Please provide literature for this speculation.**

Response: Relevant literatures have been added in this revision per your kind suggestions, which is shown as follows:

*"In contrast, the pronounced daytime rainfall dominance in South China may arise from the interaction between enhanced onshore monsoonal flows and terrain (Bai et al., 2020), sea breeze fronts and cold pool (Chen et al., 2016)"*

*References*

*Bai, L., Chen, G., Huang, Y., and Meng, Z.: Convection initiation at a coastal rainfall hotspot in South China: Synoptic patterns and orographic effects, J. Geophys. Res. Atmos., 126(24), e2021JD034642, doi: 10.1029/2021JD034642, 2021.*

*Chen, X., F. Zhang, and K. Zhao: Diurnal Variations of the Land–Sea Breeze and Its Related Precipitation over South China. J. Atmos. Sci., 73, 4793–4815, doi:10.1175/JAS-D-16-0106.1, 2016.*

**13. L215–216: "52.5%" and "47.5%" is the relative contribution rate of frequency, but not the frequency of nocturnal rainfall or daytime rainfall. Please clarify the frequency at national scale that is quite confusing in current statement.**

Response: We have revised the sentence to explicitly clarify that these values represent the relative proportion of occurrence frequency.

**14. L222: "displayed" -> "displays".**

Response: Amended as suggested.

**15. L223–226: I think the more intense nocturnal rainfall should be attributed to the absolute magnitude of nocturnal LLJs. It seems unreasonable to relate this intense rainfall with the enhancement of nocturnal LLJs relative to daytime LLJs.**

Response: We fully agree with the reviewer's opinion. It is inaccurate to relate intense rainfall with the enhancement of nocturnal LLJs relative to daytime LLJs. We have revised the text to explicitly emphasize that it is the high occurrence frequency and strong absolute intensity of nocturnal LLJs that spatially correspond to the intense nocturnal rainfall, treating the diurnal difference primarily as background context:

*"Nocturnal LLJs activities occurred more frequently, with an overall occurrence frequency increase of nearly 18% (Figs. 3a and 3e). Spatially, the regions exhibiting pronounced jet activity and high absolute wind speeds were collocated with those experiencing intense nocturnal rainfall, particularly over northern and eastern regions. Vertically, these jets manifest as intensified LLJ core concentrated below 1 km AGL (Figs. 3f and 3g)."*

**16. Figure 2: It seems (a)–(f) here are unnecessary to be shown since they are not cited in the text. It would be better to add relevant description for these subplots, or delete them.**

Response: Per your kind suggestions, relevant descriptions for subplots of a-f have been added.

**17. Figure 3d–h: Why only four directions for LLJs, but not eight directions?**

Response: we have re-analyzed the data using an eight-direction classification (N, NE, E, SE, S, SW, W, NW) to better capture the subtle variations in the low-level flow. The revised figure is as follows:

[Figure]

Figure 3. (a–d) Spatial distribution of occurrence frequency, height, strength, and the dominant wind direction of LLJs observed by 31 RWP stations during April–October from 2023 to 2024 in the daytime. (e–h) Same as (a–d), but in the nighttime.

**18. L247: Please consider deleting "Therefore". There is no causal relationship between the two sentences.**

Response: Deleted as suggested.

**19. L251–253: The statement of "nearly 45.0%" cannot be well supported by the pie charts in Fig. 6.**

Response: Thanks for the correction. We have removed the citation "(see the pie charts in Fig. 6)" in the revised manuscript to avoid confusion. Furthermore, we have

thoroughly proofread the entire manuscript to ensure strict consistency between all textual descriptions and their corresponding figures.

**20. Figure 7: Do the blue line denote the accumulated LLJ frequency over 0–3 km latitude?**

Response: Yes. We have clearly stated this in the figure caption.

**21. L289: Consider deleting "However".**

Response: Amended as suggested.

**22. Figure 8c: The maximum wind speeds are lower than 10 m/s. I wonder how they satisfy the LLJ definition in section 2.**

Response: The LLJ definition ($\geq$10 m/s) is applied to identify individual LLJ profile. However, Figure 8 presents a average result of wind profiles within 2 hours preceding to all rainfall events. Due to the variability and difference in the exact height and timing of the jet core across different events, the averaging process inevitably "smooths out" the peak intensities. Consequently, the mean maximum wind speed naturally appears lower than the threshold required for individual detection. This is an expected statistical outcome and a standard statistical characteristic of composite analysis in synoptic meteorology (e.g., Wei et al., 2013; Li et al., 2024).

**23. L322–324: The double LLJs in Du and Chen (2019) are at nearly 950 hPa (BLJ) and 700 hPa (SLLJ). How the bimodal peaks here correspond to their double LLJs?**

Response: The concept of "double LLJ" fundamentally refers to the coexistence of a BLJ and a SLLJ. While Du and Chen (2019) utilized specific pressure levels (e.g., 950 hPa and 700 hPa) as diagnostic proxies to represent these features in large-scale synoptic analysis, the underlying physical definitions are based on vertical height ranges rather than fixed pressure surfaces. According to the standard definition established in Du et al. (2012), BLJs are characterized by jet cores occurring below 1 km, whereas SLLJs are defined by cores within the 1–3 km layer. Therefore, the

bimodal peaks observed in our study correspond precisely to these two distinct height categories, demonstrating that our vertical profile results are fully consistent with the "double LLJ" structure in South China described in previous literature.

Nonetheless, the phrasing used here to reference the literature could be misleading. Accordingly, we have amended the text.

**24. L335: It should be "under 1.7 km"?**

Response: We thank the reviewer for this careful check. After a thorough re-examination of the Figure 9, we have corrected the value to "nearly 1.5 km" in the revised manuscript to ensure accuracy.

**25. L360: The choice of 1 hour instead of 2 hours used before should be further explained.**

Response: Our choice of the 1-hour window for thermodynamic analysis was driven by two considerations:

First, large-scale thermodynamic fields (e.g., temperature, moisture) generally evolve more slowly than the fine-scale kinematic features of LLJs. We examined the conditions at 2 hours prior, 1 hour prior, and the rainfall onset time, and found that the thermodynamic patterns remained largely consistent across these intervals.

Second, the 1-hour preceding window was selected as the most representative snapshot of the "immediate pre-convective environment" that directly supports the subsequent rainfall. While Section 3.2 focused on the process of LLJ evolution (requiring a 2-hour dynamic window), Section 3.3 focuses on the background state, for which the 1-hour snapshot is sufficient and appropriate.

The above content has been supplemented and revised in the original text.

**26. L363: "MFC" stands for?**

Response: Apologize for the omission of full name for the acronym that appears in the first time. Actually, "MFC" stands for moisture flux convergence, which has been added in this revised manuscript.

**27. L365–387: The authors are encouraged to add more specific citation of Fig. 10 and Fig. 11 for these statements, for the convenience of reader.**

Response: Amended as suggested.

**28. L390–391: The physical meanings of "LLJ index" and "VWS" here are unclear, especially "LLJ index". Please try to clarify these terms.**

Response: Agreed. Per your kind suggestion, we have revised the manuscript to explicitly state both the calculation procedures and the reason for selecting these two variables.

The LLJ index and vertical wind shear (VWS) were used to quantify the minute-scale dynamical coupling between jet intensity and vertical structure immediately prior to rainfall onset, which provides more integrated insights than analyzing wind profiles in isolation.

For LLJ index, it is calculated as the maximum wind speed below 3 km divided by the height where the wind speed first exceeds 10 m s⁻¹ (Liu et al., 2003). This index compactly links jet intensity and the vertical position where the jet becomes established. A rapid rise in LLJ index indicates a stronger jet and/or a lower occurrence height, quantitatively reflects the downward extension and pulsing intensity of the LLJ. And its magnitude has been shown to be positively correlated with subsequent precipitation intensity 1-2 hour later, providing indicative value for nowcasting.

For VWS, it is calculated as the wind-speed difference between the near-surface level and the jet level divided by the jet height (Wei et al., 2014), which measures the bulk shear from the surface to the jet layer. Enhanced VWS provides a more favorable environment for lifting and organization or intensification of convection.

The above-mentioned response and clarification have been well incorporated in this revised manuscript in the section 3.3.

**29. L403: weaker increases in surface θe?**

Response: Apologize for the incorrect description. We have made the revisions in our manuscript as follows:

"*During Phase 4 in ROI-4, under the favorably thermal environments ($\theta_e >$ 346 K), LLJ_HR events showed a two-stage dynamic intensification. Initially, the LLJ index surged, while the VWS and jet intensity reached synchronous secondary peaks at -72 min. In the second stage, VWS increased rapidly by ~0.9 (Fig.12d), and the LLJ index maintained an overall upward trend, peaking immediately prior to onset due to the surging jet.*"

**30. L279–318, L365–447: The presentation of the results across different regimes appear to be burdensome for readers. Please consider reorganizing these texts to highlight the similarities and differences among the regimes.**

Response: Amended as suggested.